

# Resilience of BST-2/Tetherin structure to single amino acid substitutions

Ian R. Roy[1], Camden K. Sutton[2] and Christopher E. Berndsen[3,4]

[1] Department of Health Sciences, James Madison University, Harrisonburg, VA, United States of America
[2] Department of Kinesiology, James Madison University, Harrisonburg, VA, United States of America
[3] Department of Chemistry and Biochemistry, James Madison University, Harrisonburg, VA, United States of America
[4] Center for Genome and Metagenome Studies, James Madison University, Harrisonburg, VA, United States of America

## ABSTRACT

Human tetherin, also known as BST-2 or CD317, is a dimeric, extracellular membrane-bound protein that consists of N and C terminal membrane anchors connected by an extracellular domain. BST-2 is involved in binding enveloped viruses, such as HIV, and inhibiting viral release in addition to a role in NF-kB signaling. Viral tethering by tetherin can be disrupted by the interaction with Vpu in HIV-1 in addition to other viral proteins. The structural mechanism of tetherin function is not clear and the effects of human tetherin mutations identified by sequencing consortiums are not known. To address this gap in the knowledge, we used data from the Ensembl database to construct and model known human missense mutations within the ectodomain to investigate how the structure of the ectodomain influences function. From the data, we identified an island of sequence stability within the ectodomain, which corresponds to a functionally and structurally important region identified in previous biochemical and biophysical studies. Most of the modeled mutations had little effect on the structure of the dimer and the coiled-coil, suggesting that the coiled-coil compensates for changes in primary structure. Thus, many of the functional defects observed in previous studies may not be due to changes in tetherin structure, but rather, due to in changes in protein-protein interactions or in aspects of tetherin not currently understood. The lack of structural effects by mutations known to decrease function further illustrates the need for more study of the structure-function connection for this system. Finally, apparent flexibility in tetherin sequence may allow for greater anti-viral activities with a larger number of viruses by reducing specific interactions with anti-tetherin proteins, while maintaining virus restriction.

Corresponding author
Christopher E. Berndsen,
berndsce@jmu.edu

## INTRODUCTION

Tetherin (BST-2, CD-317) is a membrane bound protein involved in a non-specific, immune response to enveloped viruses, such as HIV-1 and Ebola (*Neil, Zang & Bieniasz, 2008*; *Van Damme et al., 2008*; *Neil, 2013*; *Sauter, 2014*). During viral budding, one of the transmembrane anchors, usually the C-terminal anchor, becomes embedded in the virion

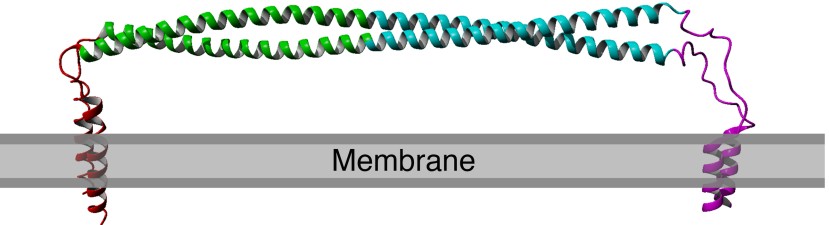

**Figure 1** **Structure of human tetherin bound to the lipid membrane. Membrane anchors are shown in red (N) or magenta (C).** The coiled-coil region within the ectodomain is shown in cyan. Image derived from the model produced by *Ozcan & Berndsen, (2017)*. The C-terminal anchor is shown as an alpha helix, however there is evidence that in some forms of tetherin, this may be a GPI anchor (*Kupzig et al., 2003*; *Rollason et al., 2007*).

particle membrane and prevents the diffusion of the virion away from the host cell (*Perez-Caballero et al., 2009*; *Venkatesh & Bieniasz, 2013*). Viruses have developed numerous methods to evade tetherin and this is an area of intense study (*Sauter, 2014*; *McNatt, Zang & Bieniasz, 2013*). Additionally, tetherin is hypothesized to cause an inflammatory response via activation of the NF-kB pathway and may be involved in cell migration (*Galão et al., 2012*; *Hotter, Sauter & Kirchhoff, 2013*). The structural basis for these functions is not fully clear.

Structurally, tetherin is a homo-dimer consisting of an N-terminal transmembrane helix and a C-terminal membrane anchor bridged by an alpha helical ectodomain (Fig. 1) (*Kupzig et al., 2003*; *Rollason et al., 2007*; *Andrew, Miyagi & Strebel, 2011*). There are three disulfide bonds within the N-terminal portion of the ectodomain that connect to the two monomers of the dimer and appear to enhance the tensile strength of the protein (*Du Pont et al., 2016*). Computational simulation of viral tethering has suggested sequence encoded weaknesses in the C-terminal part of ectodomain that enable bending and flexing of the ectodomain during the transition from membrane bound to bridging the virus and host cell (*Ozcan & Berndsen, 2017*). Despite these specific features, upwards of half of the coiled-coil region can be removed without significant loss of function (*Andrew et al., 2012*).

A few previous studies of tetherin structure and function have used either scanning mutagenesis or analyzed a subset of known human mutations (*Welbourn et al., 2015*; *Hammonds et al., 2012*; *Yang et al., 2010*). Cysteine scanning mutagenesis within the ectodomain showed that mutations in the coiled-coil region altered tetherin dimerization and interfered with function, while cysteine mutations outside of this region were often functional (*Welbourn et al., 2015*). The new cysteines changed the orientation of the helices and lead to changes in the folded state of tetherin. Alanine scanning mutagenesis via changing of the sequence four alanines at a time showed two areas of the tetherin ectodomain were sensitive to quadruple alanine mutation, however most mutations had no apparent effect (*Hammonds et al., 2012*). The likely cause of functional defects was a change in protein localization (*Hammonds et al., 2012*). Mutations that reversed the charge of three to six surface amino acids in the N-terminal portion of the ectodomain did reduce viral tethering, however whether this was due to alterations in structure or virus binding

was unclear (*Yang et al., 2010*). Deletions within the ectodomain also showed defects in viral tethering (*Yang et al., 2010*). However, a later study by Andrew and coworkers showed that deletions in the ectodomain that alter the register of the alpha helix are detrimental while mutants with truncations that maintain the orientation of the helices are functional (*Andrew et al., 2012*). These previous examples suggest that the tetherin sequence is amenable to change as long as the alpha helical structure of the ectodomain is maintained.

In order to more fully understand the structure-function connection in tetherin, we analyzed structural models of tetherin containing one of 78 single nucleotide polymorphisms listed in the Ensembl database (*Aken et al., 2017*). In all cases, the mutations did not affect the global structure of tetherin and instead affected local structure primarily at the ends of the ectodomain. Mutations that occurred at the interface of the tetherin monomers induced greater changes in protein dynamics than those that occur on the solvent-facing surface of the ectodomain. These data further support the idea that mutations in tetherin are tolerated provided that the alpha helices in the ectodomain remain intact.

## METHODS

### Production of tetherin mutants

Tetherin mutants were generated from the membrane bound model of tetherin described by *Ozcan & Berndsen (2017)* using the list of human mutations found in Ensembl as of August 2018 (*Aken et al., 2017*). We wrote a YASARA macro to individually swap amino acids and minimally energy minimize the resulting structure. All models were equilibrated for 10 to 20 nanoseconds in explicit water solvent and a DOPC bilayer using an AMBER14 forcefield within YASARA structure (*Case et al., 2014*; *Krieger & Vriend, 2015*; *Krieger et al., 2009*). Simulation conditions were 298 K, pH 7.4, 0.9% mass fraction NaCl, at a density of 0.997 g/mL, with a 12 Angstrom non-bonded interaction cutoff, and a time step of 2.5 femtoseconds.

### Analysis of models

Mutant simulations were analyzed using existing macros within YASARA to calculate simulation energies, root mean square fluctuation (RMSF), and the secondary structure over time. These macros were run on the complete data set. Data for each mutation were summarized using Rmarkdown. Root mean square deviation values were calculated using the Align function in YASARA structure. Global analysis of data was performed in R using the components within the *tidyverse* package and the *cluster* package (*Ross, Wickham & Robinson, 2017*; *Wickham, 2016*; *Maechler et al., 2013*). Ridgeline plots and some figures were made using the *ggridges* and *cowplot* packages of Claus Wilke (*Wilke, 2018a*; *Wilke, 2018b*).

## RESULTS

### Human missense mutations are absent from beginning of coiled-coil

The structure of tetherin has been determined however the connection between the crucial functions of tetherin and this structure is less clear (*Swiecki et al., 2011*; *Hinz et*

al., 2010; Schubert et al., 2010; Yang et al., 2010). The primary sequence of tetherin is not well-conserved across species, although the general organization appears to be (*Heusinger et al., 2015*; *Blanco-Melo, Venkatesh & Bieniasz, 2016*). Areas of structural or functional importance generally have a conserved protein sequence. Given the lack of apparent sequence conservation between species, it is difficult to predict the important structural features of tetherin. We decided to compare sequences within the human species to identify areas of amino acid stability and therefore relative importance. We retrieved variation data from the Ensembl database containing the missense mutations within the ectodomain of tetherin (Table S1). We focused on the mutations within the helical portion of the ectodomain because there is better, high quality structural information on this region. Mapping the amino acid locations of SNPs showed that there was a lower density of mutations between residues 93 and 117 (Fig. 2A). We note that synonymous mutations, which do not affect the amino acid sequence, still occur in this region, but amino acid changes appear to be lacking (Fig. 2A). Although the data set is limited in scope, the information suggests that amino acids in this region, which includes one disulfide and the beginning of the coiled-coil region of the ectodomain, are important for tetherin structure or function. In addition to a general position search, we mapped the mutations on the 3MQB structure and observed that 66 of 78 mutations appear to occur on the exterior of the protein ("outside") rather than at the interface of the tetherin dimer ("inside") (Fig. 2B) (*Yang et al., 2010*). Moreover, the frequency of many of these outside mutations is higher than for inside mutations (Table S1). The simplest interpretation of this distribution of mutations is that the solvent and membrane facing surfaces of the protein are more tolerant of single amino acid changes. This distribution supports the previously demonstrated role of dimerization in viral tethering and stability, although dimerization is not entirely essential for tethering of all viruses (*Andrew et al., 2009*; *Sakuma, Sakurai & Yasuda, 2009*; *Schubert et al., 2010*; *Welbourn et al., 2015*; *Du Pont et al., 2016*). The 78 mutations occur at 59 unique sites within the protein and some positions have several polymorphisms (summarized in Table S1). We also observed that the areas of highest mutation density were in the N-terminal portion of the ectodomain, which has been suggested to be more flexible and is stabilized by the inter-molecular disulfides (*Welbourn et al., 2015*; *Du Pont et al., 2016*), or at the C-terminal end of the coiled-coil region, which in structural simulations unwinds relatively easily as it contains a non-canonical heptad (*Andrew et al., 2012*; *Du Pont et al., 2016*; *Ozcan & Berndsen, 2017*). Therefore the locations of SNPs align well with previous functional and structural data on tetherin. However, the potential effects of these mutations on tetherin structure are unclear.

## Missense mutations do not alter overall structure

To determine the association between structure and function in tetherin mutants, we computationally made point mutations in the previously described full-length model of tetherin at positions between 49 and 163, where the structure of tetherin appears mostly helical (*Ozcan & Berndsen, 2017*). All models equilibrated within 10 to 20 ns of simulation. We then compared the root mean square deviation, root mean square fluctuation, and secondary structure for all mutations to a similar simulation on a model lacking mutations
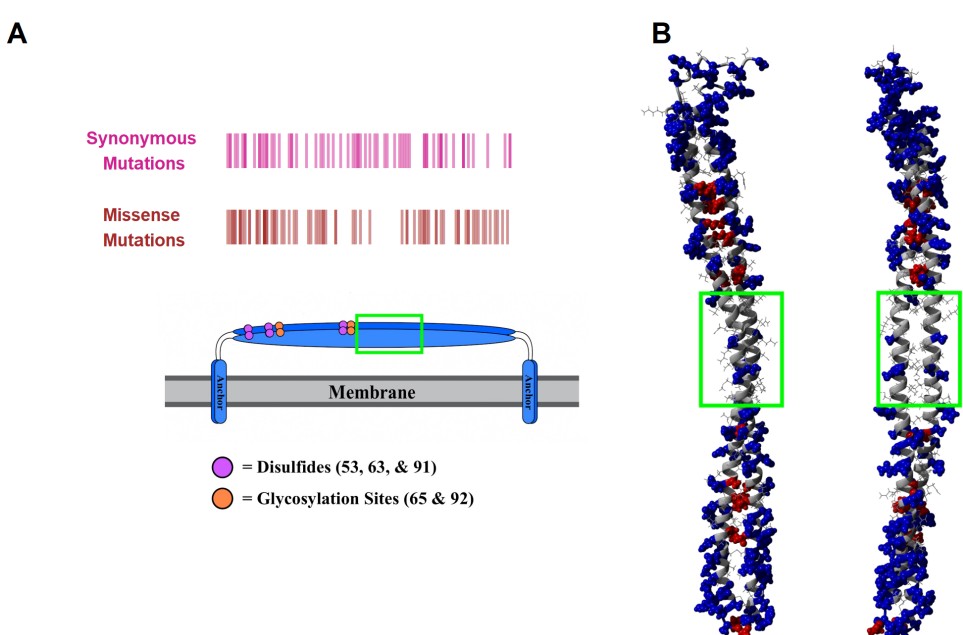

**Figure 2** **Human mutations within tetherin.** (A) Cartoon showing domain organization and location of synonymous (purple) and missense mutations (red) in tetherin (B) Model showing the location of the mutants on the inside (red) or outside (blue) of the tetherin ectodomain. The green box in (A) and (B) indicates the location of amino acids 93–117.

to identify structural and dynamic changes due to the mutation. The root mean square deviation (RMSD) is a measure of the differences in the 3-D position of one or more elements of the ectodomain. We calculated the RMSD of the average structure from each mutant simulation from alignment of the ectodomain alone with a wild-type tetherin that had also been equilibrated. The data were then plotted as RMSD vs. amino acid number as shown in Fig. 3. Any large RMSD values would suggest a strong effect of the mutation on the overall tetherin structure. We observed that there were distinctly large RMSD values across the ectodomain mutations and no single mutation disrupted the entire coiled-coil (Fig. 3). These data suggest that mutations in tetherin do not cause major changes to the helical structure.

## Mutations alter local tetherin structure

Single changes to the amino acid sequence may not cause drastic changes to protein structure. Therefore, we next analyzed our simulations for local changes in helical structure. Based on the distance between hydrogen bond donors and acceptors, amino acid positions were binned into types of secondary structure at 400 steps throughout the simulation. Representative plots of secondary structure across the simulation are shown in Fig. 4. In addition, we simulated a point mutation known to affect the structure of tetherin, Gly118Cys, as a positive control for structural disruptions (*Welbourn et al., 2015*). The wild-type simulation shows a stable helical structure which is matched in behavior by the Asp129Glu and Asp103Asn mutations. The postive control, Gly118Cys, shows a chunk

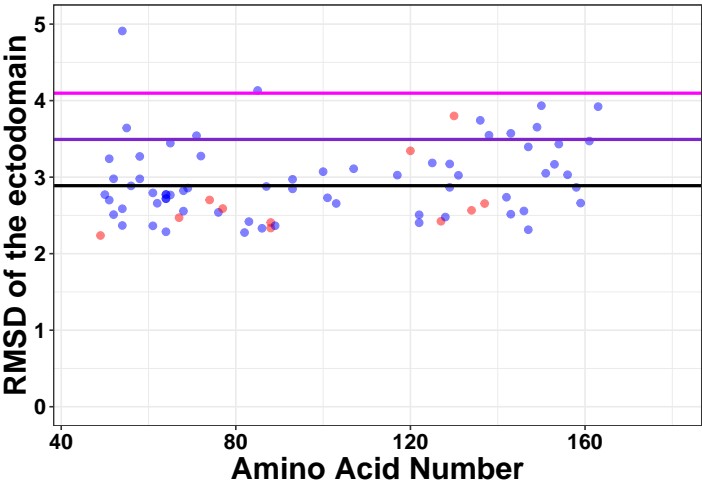

**Figure 3** **No large changes in structure induced by mutations as shown by the root mean square deviation plotted against amino acid number.** Red points are those positions that are found on the inside of the tetherin dimer, while blue points are those positions on the outside of the dimer. Black, purple, and magenta lines show the mean, mean + one and mean + two standard deviations of the RMSD values respectively. Mutations at positions C-terminal to amino acid 163 were not analyzed because this region appears unstructured in crystal structures (*Yang et al., 2010*; *Hinz et al., 2010*; *Swiecki et al., 2011*).

of 4 amino acids which are in a coiled structure, but the effects are limited to this region. Similarly, mutations Arg64Pro, Ala100Pro and Ile120Phe showed distinct regions where the secondary structure is altered compared to the wild-type as shown in Fig. 4A. Proline is well-known to be disruptive to alpha helices because it is unable to form the hydrogen bonds characteristic of this secondary structure (*Pace & Scholtz, 1998*). Position 64 within tetherin is the most diversely mutated position in the data that we analyzed with 5 SNPs but only the Arg64Pro had any effects on structure (Table S1). Phenylalanine is aromatic and occupies a larger volume than isoleucine suggesting the local secondary structure changes are due to the alterations in the interface between the two helices. We note that none of the changes in secondary structure spread along the coil. Instead, cracks in the helices stayed localized to a single area within the ectodomain, supporting the biochemical and computational findings of *Du Pont et al. (2016)*. We next analyzed where the changes in structure were localized and plotted the number of times an amino acid adopted a different environment versus position (Fig. 4B). The effects on secondary structure appear primarily at the ends of the coil, which are known to be unstructured, rather than proximal to the site of substitution except for a few mutations (Fig. 4B) (*Swiecki et al., 2011*). These data suggest that the potential disruptive effects of most mutations are minimized by the coiled-coil structure of tetherin.

While most mutations had minimal effect on the helical structure of tetherin, three mutations Arg64Pro, Ala100Pro, and Ile120Phe did disrupt the helices. Arg64Pro and Ala100Pro induced alterations in secondary structure near to the site of the polymorphism. In neither of these instances do the alterations in structure include more than two consecutive amino acids. Ile120Phe however induces a disruption in secondary structure

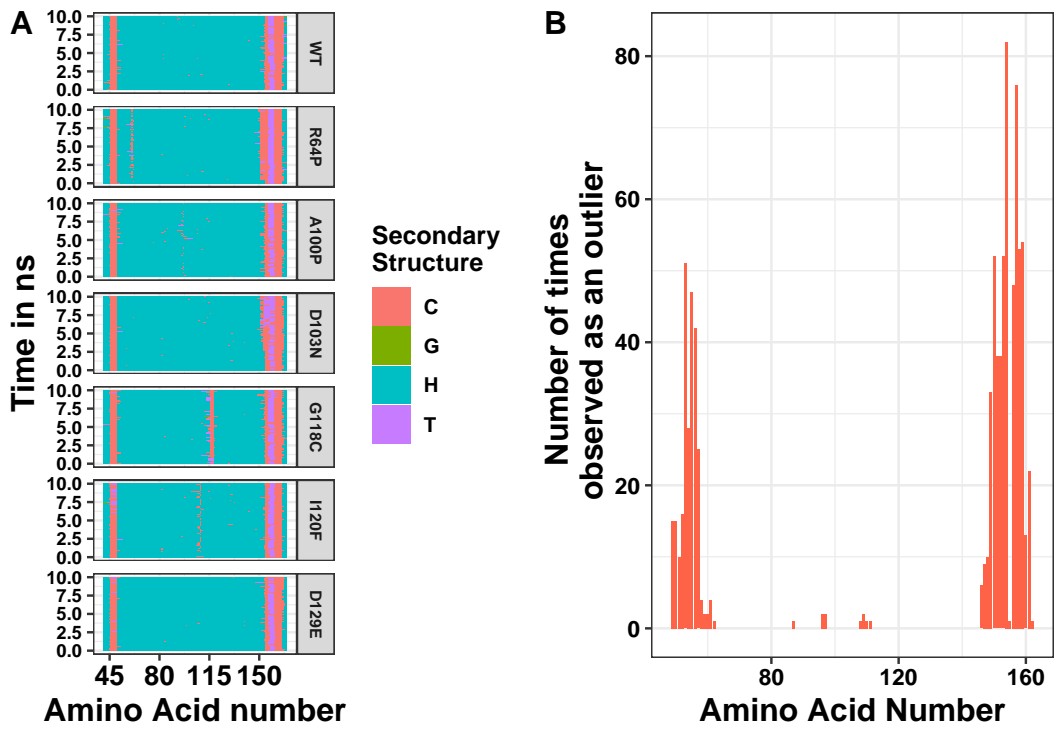

**Figure 4** (A) Secondary structure of tetherin mutant simulations over time for Arg64Pro, Ala100Pro, Gly118Cys, Ile120Phe, Asp129Glu, and Asp103Asn. Secondary structures were assigned by YASARA based on hydrogen bonding distances and psi angles at 25 picosecond intervals. 'H' stands for alpha helix, 'T' for turn, 'C' for coil, and 'G' for 3/10 helix. (B) Plot showing the cumulative number of times that the secondary structure was perturbed at each location for all simulations. Outliers were identified as sites that had distinct secondary structure more than two standard deviations above the average secondary structure value at that site across all simulations. A value of one means that the amino acid appeared as an outlier in either molecule of the dimer in 1 simulation.

that is not directly adjacent to the mutation site and is maintained throughout the simulation. For Ala100Pro mutation this disruption in secondary structure was reproduced while for Ile120Phe, the disruption in secondary structure was inconsistent (Fig. S1). Visualization of the energy minimum structure from the simulation for both the wild-type and Ile120Phe structure shows that Ile120 is an interior amino acid and the side chains could form a van der Waals contact (Fig. 5A). However, in the structure of the Phe substitution, the side chain is displaced from the interface (Fig. 5A). Further comparison of the distance between the tetherin monomers and at sites along the helix shows that the Phe substitution noticeably changes the distribution of measurements on either side of the mutation over a region extending at least 56 amino acids in length (Fig. 5B). The alteration in the amino acid distances appeared consistently in replicate simulations of this mutation despite the lack of effect on secondary structure in some simulations (Fig. S2). In comparison, the Ala100Pro mutation does not alter distribution of distances except for slight differences adjacent to the mutation site (Fig. 5C). Observation of the individual measurement sites over time show that amino acids in the region of 105 to 123 quickly adjust to a new distance

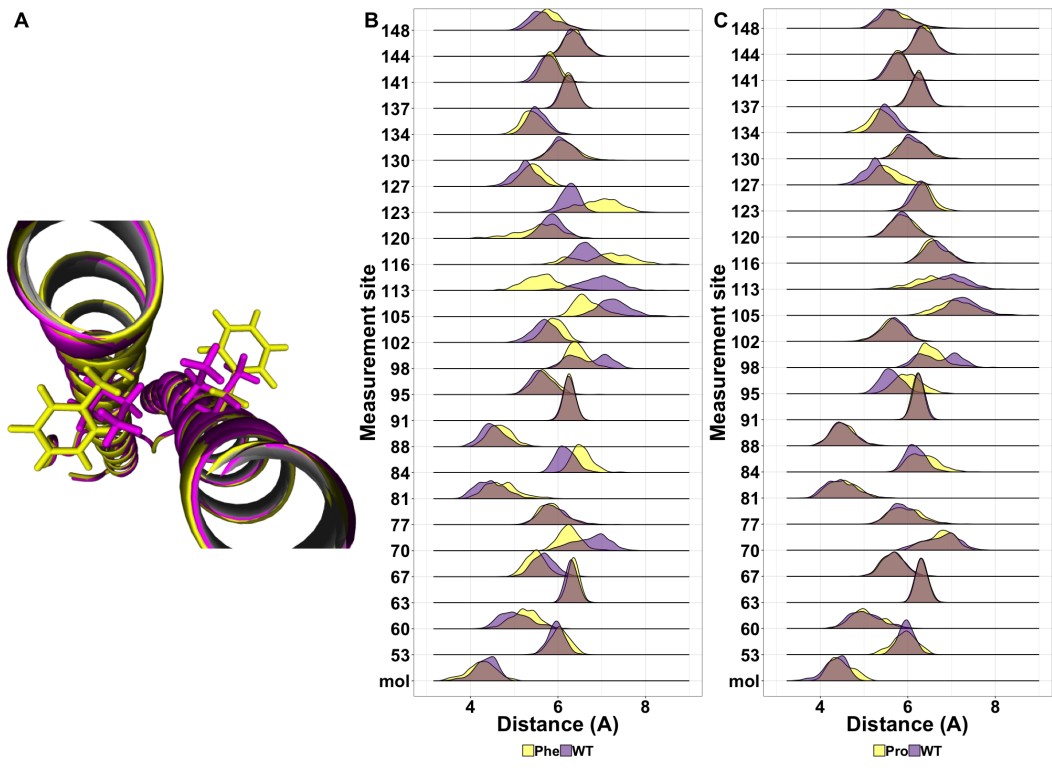

**Figure 5** (A) Differences in side chain position between the wild-type Isoleucine (purple) and the mutation Phenylalanine (yellow) (B) Ridgeline plot showing the range of distances between the A and B molecules (mol) or the Calpha at the indicated positions comparing the Ile120Phe simulation (yellow) and the wild-type simulation (purple). Overlap is shown in pink. (C) Ridgeline plot showing the range of distances between the A and B molecules (mol) or the Calpha at the indicated positions comparing the Ala100Pro simulation (yellow) and the wild-type simulation (purple). Overlap between the two conditions is indicated by grey coloring.

followed by slower adjustments at positions 81, 84 and 148 (Fig. S3). This behavior is not observed in the wild-type simulations (Fig. S3). These data suggest that the structure of tetherin can tolerate many changes in the sequence and that for most changes that disrupt the helical structure there is limited spreading of breaks.

## Effect of mutations on ectodomain dynamics

Protein structure is not just based on the static arrangement of amino acids, but the ensemble of structures that the protein can adopt, which is sometimes referred to as protein dynamics. To quantify dynamics, we analyzed the root mean square fluctuation (RMSF), which describes the amount of movement each amino acid undergoes during the simulation. We then correlated the per residue RMSF values between each mutation simulation and the wild-type simulation. We observed moderate correlations at most positions but wanted additional methods to analyze the shape of the cluster. Using the central point in the cluster would allow for easier visualization of how each mutation affects the global dynamics of the system and allow for easier comparison of the effects of mutations. Therefore, we next calculated the mean and medoid centers of the RMSF

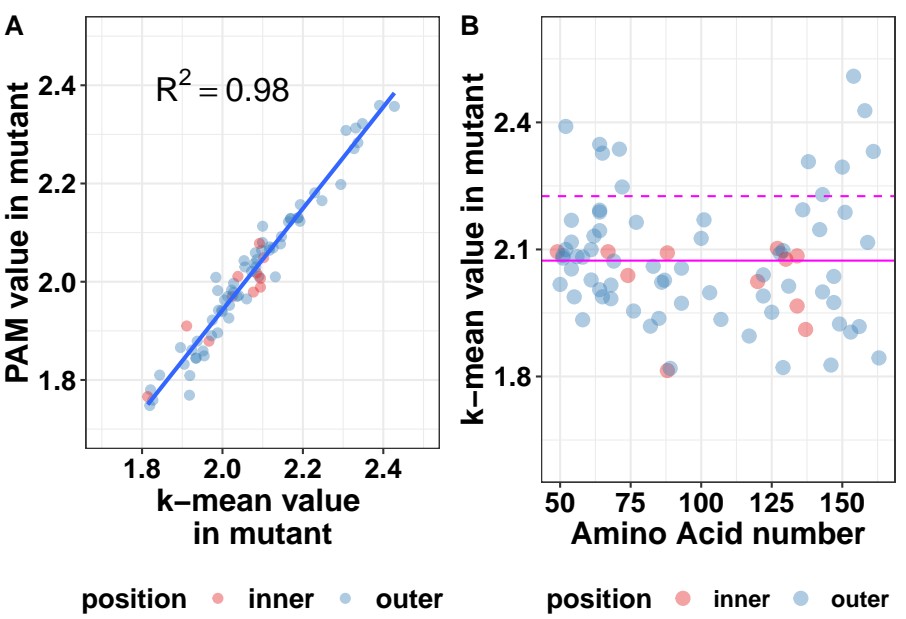

**Figure 6** **Mutation effects on protein dynamics (A) correlation of k-means and PAM values (B) scatterplot of k-mean values at each amino acid position.** Magenta lines indicate the mean value (solid) and the mean plus one standard deviation (dashed) of the k-means. Mutation effects on protein dynamics (A) correlation of k-means and PAM values (B) scatterplot of k-mean values at each amino acid position.

clusters. The data for the medoid and mean cluster centers in the mutants correlated well suggesting that our data are normally distributed (Fig. 6). We plotted all the data determined by the k-means method at each amino acid position to determine if there were any trends or clustering. As seen in the right pane of Fig. 6, mutations at the ends of the ectodomain tended to have larger k-mean values suggesting larger RMSF values and more dynamics in the simulation. Mutations on the interface of the two helices did not increase dynamics relative to the outside mutations. These data are consistent with our previous finding that disruptions to secondary structure occurred preferentially toward the termini of the ectodomain and that the ectodomain can tolerate some changes to the structure.

## Comparison with Tetherin mutations with known functional defects

Given the general lack of effect of point mutations on the structure of tetherin, we next looked at known non-functional mutations described previously in the literature (Table S2). We have shown that cysteine mutations or truncations that alter the orientation of the helices resulting in a poor interface are non-functional (*Welbourn et al., 2015*; *Andrew et al., 2012*). However, others have shown multiple alanine or charge swap mutations can also reduce viral tethering (*Yang et al., 2010*; *Hammonds et al., 2012*). Yang and coworkers proposed that swapping the charge of amino acids in the N-terminus altered the interaction of tetherin with the lipid membrane. In Fig. 7A, we suggest that this is unlikely because the mutation sites are not on a single face of the coiled-coil and not all sites interact with the membrane in our model. However, we cannot rule out a mechanism where the orientation of the helices relative to the membrane are maintained via interactions between these
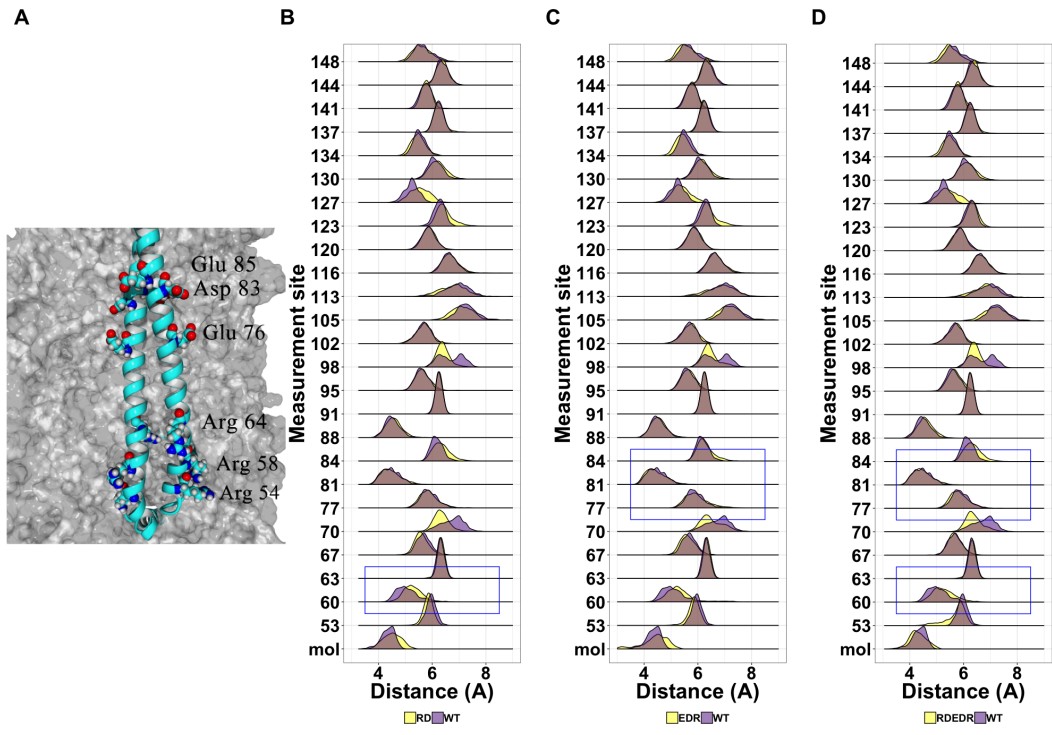

**Figure 7** (A) Position of the charge swap mutations within the tetherin coiled-coil (B)–(D) Ridgeline plots showing the range of distances between the A and B molecules (mol) or the Calpha at the indicated positions comparing the charge swap mutant simulations (yellow) and the wild-type simulation (purple). Overlap is shown in grey A blue box indicates the area where these mutants are within the sequence. (B) Arg54, Arg 58, Arg64 to aspartate. (C) Glu76, Asp83, Glu85 to arginine. (D) Both sets of mutants.

amino acids and the lipid heads. Simulations show a relatively stable distance between the coils with few disruptions outside of the sites of mutation (Figs. 7B–7C).

We next analyzed simulations containing the tetra-alanine and tetra-serine mutations of tetherin by *Hammonds et al. (2012)*. As with the charge swap mutations, the tetra-alanine mutations show essentially no structural changes (Fig. 8). In contrast, the tetra-serine substitution showed large but localized changes in structure as assessed by distance measurements (Fig. 8C). Given that the tetra-serine mutation sites are all at the interface of the coiled-coil, these data further link alterations of the coiled-coil interface to defects in function (*Welbourn et al., 2015*; *Andrew et al., 2012*; *Hammonds et al., 2012*; *Yang et al., 2010*; *Du Pont et al., 2016*). Additionally, these results match up to our results with the single point mutations showing limited structural effects induced by mutation.

## DISCUSSION

The connection between tetherin structure and function continues to be a significant question in the field. Putative orthologs of tetherin show little sequence similarity and there are only structures for two orthologs (*Hinz et al., 2010*; *Swiecki et al., 2011*; *Schubert et al., 2010*; *Yang et al., 2010*; *Heusinger et al., 2015*; *Blanco-Melo, Venkatesh & Bieniasz,*

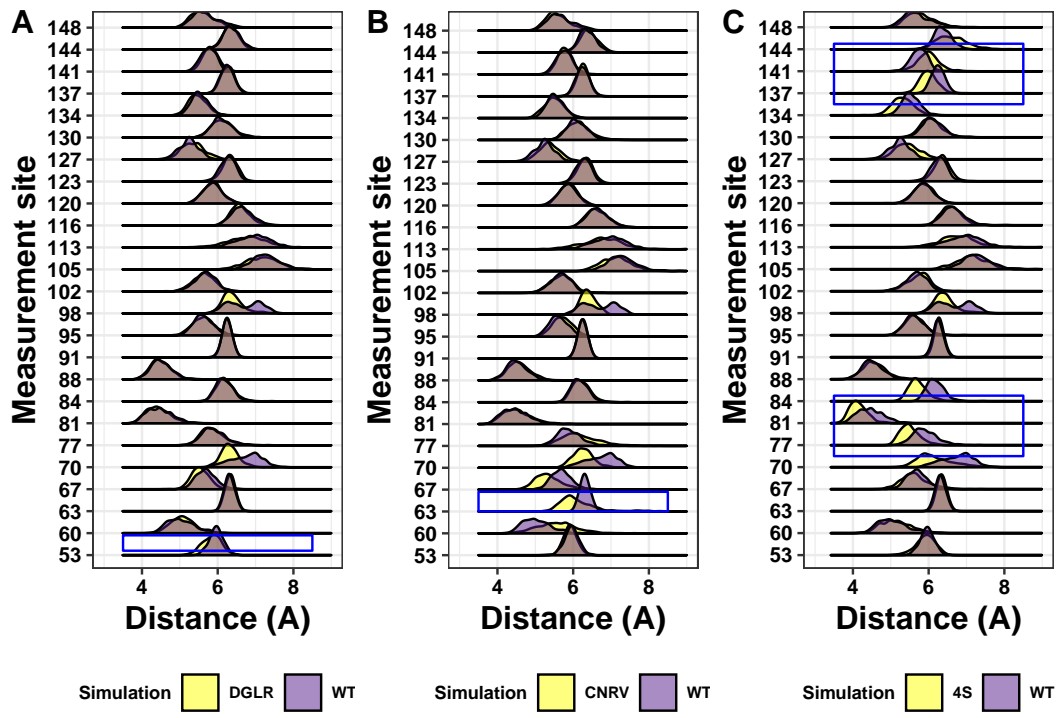

**Figure 8** **Ridgeline plots showing the range of distances between the Calpha at the indicated positions comparing the tetra-substituted simulation (yellow) and the wild-type simulation (purple).** Overlap is shown in grey. Blue boxes indicate the region where the mutation occurs within the protein. (A) Asp55, Gly 56, Leu57, Arg58 to alanine (B) Cys63, Asn64, Arg65, Val66 to alanine. (C) Val74, Val84, Leu137, Leu144 to serine.

*2016*). In order to gain more insight into the structural pliability of tetherin, we analyzed the location and effects of tetherin mutations found in the human genome. While the data set that we analyzed only includes data from those genome studies that deposit in Ensembl, this is the most comprehensive set of genomes available. Initial analysis showed a relative lack of missense mutations between amino acids 93 and 117. This section of tetherin includes one cysteine which can form a disulfide, a glycosylation site, and also the first three heptads of the coiled-coil (*Andrew et al., 2009*; *Andrew et al., 2012*). While there are synonymous sequence changes in the DNA sequence along the entire length of the protein, this section lacks mutations that change the protein sequence. Our analysis of these mutations supports previous work showing this region as being important for flexibility or hinge motions (*Hinz et al., 2010*; *Yang et al., 2010*; *Andrew et al., 2012*; *Ozcan & Berndsen, 2017*). Collectively, these data suggest that the flexibility of this region is tuned for a specific structure and/or function. Previous work suggested that this purpose could be to "break" in a controlled fashion during the transition from the membrane bound to the cell-virus bridging form (*Ozcan & Berndsen, 2017*). However, more information is needed on the molecular functions of tetherin to explore and test this idea fully.

Mapping of the mutations onto the structure of tetherin showed that most mutations occur on amino acids on the exterior of the helices rather than at positions at the dimer

interface (Fig. 2) (*Yang et al., 2010*). The importance of tetherin dimerization for most, but not all, function is well-established, therefore fewer interior mutations was not surprising (*Andrew et al., 2009*; *Sakuma, Sakurai & Yasuda, 2009*; *Schubert et al., 2010*; *Welbourn et al., 2015*; *Du Pont et al., 2016*). Structurally, most amino acid substitutions, except glycine and proline, have similar propensities to form an alpha helix, therefore a single amino acid change regardless if it is on the inside or outside of the coil should not alter the helical structure of tetherin to a significant degree (*Pace & Scholtz, 1998*). Why then is the inside vs outside distribution interesting? Our data and analysis show that mutations on the inside of the coiled-coil occur less frequently (Table S1) and some show disruptions of the coiled-coil structure. If the mutations are likely not affecting the helical structure, this suggests the lower mutation rate of inside amino acids may be to avoid disrupting the interactions between the coils. One inside mutation that demonstrates this, Ile120Phe, showed changes, although consistently, to the helical structure of the protein (Fig. 4 and Fig. S1). We linked changes that did appear to displacement of the side chain which led to changes in the distance between amino acids in each monomer of the dimer (Fig. 5 and Figs. S2 and S3). The apparent disruption of the tetherin dimer interface and relative change in the orientation of the monomers was transmitted down the coiled-coil. These data suggest that the tetherin dimer limits disruptions to localized regions and that more drastic reorganizations of the dimer are required to induce unfolding of the structure, such as those from cysteine scanning mutagenesis performed by *Welbourn et al. (2015)*. While not all inside mutations caused drastic effects, the database where we obtained our SNP information likely does not contain many SNPs where the protein disrupted function enough to alter organism survival. Comparison of our simulations to simulations of mutations with known functional defects further demonstrated the relatively malleability of the tetherin coiled-coil. The charge swap mutations of Yang and coworkers showed minimal observable changes in structure or distance between the monomers suggesting that the functional defects in these mutant proteins are not due to changes in the structure of tetherin (Fig. 7) (*Yang et al., 2010*). The proximity of these mutations to the glycosylation sites and a region linked to trafficking of tetherin may be the source of the functional defects as shown by mutations in this region by *Yang et al. (2010)*; *Hinz et al. (2010)*; *Hammonds et al. (2012)*; *Waheed et al. (2018)*. In the region between residues 54 and 85, there are 28 missense SNPs listed in Ensembl including two at the glycosylation site at position 65 (Fig. 2A and Table S1). Only one (Arg64Pro) showed any effects on structure (Fig. 4A), further suggesting that the functional defects in this region are due to non-structural changes in the protein.

Analysis of alanine scanning or tetra-serine mutations from Hammonds and coworkers showed localized effects on the orientation of the helices (Fig. 8) (*Hammonds et al., 2012*). The most apparent was for the tetra-serine mutation, which changed four interface amino acids to serine and showed noticiable changes in the distance between the helices local to the sites of mutation (Fig. 8). The tetra-serine mutation did not however alter secondary structure as we observed for the Ile120Phe, Arg64Pro, or Ala100Pro mutations (Fig. 4). In the two regions encompassed by the tetra-serine mutations, there are a total of 10 SNPs in Ensembl, although none of these mutations showed apparent effects on Tetherin

structure. Given the similarity in effects on the orientation of the helices between the Ile120Phe mutation and the tetra-serine mutation, it is tempting to speculate that the Ile120Phe mutant likely has reduced function, however this proposal awaits experimental confirmation of the computational predictions in this present study.

In addition to the genetic stability of the middle section of tetherin, we observed that most mutations had little if any apparent effect on the overall structure of tetherin. Calculation of the RMSD between the average structure during simulations and the wild-type protein showed no trends or large changes in the structure. Supporting these data were analysis of the secondary structure at each amino acid which showed most mutations appears to affect the ends of the ectodomain, which are known to be largely unstructured (*Hinz et al., 2010*; *Andrew et al., 2012*; *Ozcan & Berndsen, 2017*; *Swiecki et al., 2011*; *Schubert et al., 2010*; *Yang et al., 2010*). Given the importance of the alpha helical structure of tetherin to tensile strength, these findings are not surprising, although certainly unexpected given the number and diversity of mutations analyzed (*Du Pont et al., 2016*; *Andrew et al., 2012*). This is most apparent in the Arg64Pro and Ala100Pro mutations, which we expected would cause drastic changes in tetherin structure and dynamics. While the proline mutation does cause local changes in secondary structure (Fig. 4), the effects of this mutation are localized to a few amino acids proximal to the mutation site. The detrimental effects of proline in helices are well-known, however the strength of the coiled-coil and the disulfides appears to prevent spreading of this "crack" in the helix (*Pace & Scholtz, 1998*; *Du Pont et al., 2016*). Previously, Sauter and coworkers analyzed the functional effects of seven rare mutations in tetherin from the human genome (*Sauter et al., 2013*). For the mutations located in the ectodomain, the Asn49Ser mutation affected NF-$\kappa$B activation, possibly due to changes in cell surface expression while the other mutations had no effects on tetherin function (*Hammonds et al., 2012*; *Sauter et al., 2013*). The Asn49Ser mutation had no apparent effects on structure in the present study suggesting that the functional or expression defects are due to another unknown aspect of tetherin.

The computational approach used by this study showed that for a limited test set most of the effects on tetherin structure were reproducible. However, the limitations of computational approaches are clear and we urge against overly broad interpretations of the data.

## CONCLUSIONS

Immediately prior to submission of this article in 2019, we searched ClinVar for disease causing mutations in tetherin and we did not find any single nucleotide polymorphisms in the ectodomain with known links to diseases or pathogenic phenotypes. These data lead to the idea that mutations are tolerated within tetherin as long as they do not affect the overall helical structure and/or dimerization. Given that tetherin has no known enzymatic functions and the interactions of tetherin with viruses like HIV-1 can rapidly adapt to immune systems, this sequence flexibility is likely advantageous as it could decrease specific interactions with proteins that antagonize tetherin anti-viral activities. The structural stability of the coiled-coil structure in the face of a changing primary structure would permit viral tethering, while allowing evasion of viral proteins.

## ACKNOWLEDGEMENTS

We thank the James Madison University Department of Chemistry and Biochemistry, Yasmeen Shorish, Kadir A. Ozcan, and the 4 –VA Organization for the resources, professional support, and models used throughout the project. We also thank the CHEM 361 students in Biochemistry 1 during Fall 2017 for their semester long work which helped shape the direction of the investigation.

### Funding

This work was supported by NSF-REU CHE-1461175 and the 4-VA organization. The funders had no role in study design, data collection and analysis, decision to publish, or preparation of the manuscript.

### Grant Disclosures

The following grant information was disclosed by the authors:
NSF-REU: CHE-1461175.
4-VA organization.

### Competing Interests

The authors declare there are no competing interests.

### Author Contributions

- Ian R. Roy, Camden K. Sutton and Christopher E. Berndsen conceived and designed the experiments, performed the experiments, analyzed the data, contributed reagents/materials/analysis tools, prepared figures and/or tables, authored or reviewed drafts of the paper, approved the final draft.

### Data Availability

    All data and code are available on the Open Science Framework: https://osf.io/hwkj4/.

### Supplemental Information

Supplemental information for this article can be found online at http://dx.doi.org/10.7717/peerj.7043#supplemental-information.

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
