# Peer review of "Resilience of BST-2/Tetherin structure to single amino acid substitutions"

_PeerJ, doi:10.7717/peerj.7043_

## Round 0.1 · original submission · Major Revisions

Please address all the critical issued raised by both reviewers and revise your manuscript accordingly.

Reviewer 1 ·

Basic reporting

The manuscript is written in a clear and comprehensible manner. Professional English is used throughout (with a few minor typos). The authors provide all background information that is required to understand the research question and the methods used. Figures are clearly labeled. As outlined in more detail below, the manuscript may benefit from a table summarizing all SNPs that were analyzed and their effects on structure/function of tetherin.

Experimental design

The research question is well defined. The methods are suitable to model the effects of SNPs on tetherin structure. One limitation of the present study is the lack of a cell biological assay analyzing the effects of the predicted structural changes on tetherin function. For example, it remains unclear whether the Ile120Phe mutation affects the antiviral activity of tetherin by disrupting the (local) secondary structure.

Validity of the findings

In most cases, the conclusions drawn by the authors are justified. The authors nicely discuss their findings in the context of previous publications that analyzed the structure-function relationship of tetherin. In some cases, the exact identity of the SNPs that were analyzed remains unclear (see below).

Additional comments

In their present study, Roy and colleagues model the effects of naturally occurring polymorphisms on the (coiled-coil) structure of the host restriction factor tetherin. They show that missense mutations are absent from a region in the N-terminal part of the coiled-coil domain. Furthermore, they convincingly demonstrate that none of the missense mutations alters the global structure of tetherin, while some mutations have (minor) effects on local tetherin structure (e.g. G64P, A100P, I120F). Unfortunately, it remains unclear whether these mutations abrogate the antiviral activity of tetherin as the study is limited to in silico analyses. I summarized a few more specific points below that should be addressed to improve the quality of the study.

Major points:
(1) A table summarizing all SNPs in the tetherin ectodomain and their effects on structure (this study) and function (restriction of virion release, NF-κB activation) as well as their frequency in the population may be helpful.

(2) Lines 109-113, 221/222, Fig. 2B: The authors highlight that “11 of 78 mutations appear to occur on the exterior” of tetherin. Fig. 2B, however, suggests that the majority of all mutation can be found on the outside (blue). Fig. 3 shows 12 residues on the inside of the dimer. Which information is correct?
What percentage of all amino acids in the ectodomain can be found on the inside/outside? Is the enrichment for mutations on the inside/outside significant?

(3) Lines 130-132: How do the authors define global disruption of the coiled coil? Can they include a mutant disrupting the structure as positive control in Fig. 3? For example, would (non-natural) mutations between residues 93 and 117 induce global changes?

(4) Fig. 3: Which mutants were selected for analysis in Fig. 3? Fig. 2A shows several missense mutations close to the C-terminus of the coiled-coil (positions 170-180), while these mutations were obviously not tested in Fig. 3. Why not?

Minor points:
(1) lines 24-26: “most mutations have little effect on structure, even those that have been previously shown to affect function. Thus, Tetherin sequence is likely less important than maintenance of the coiled-coil structure.” I fully agree with the authors that the absence of any marked effects of SNPs on tetherin structure suggests that maintenance of the coiled coil is important for tetherin function. However, this statement may also be misunderstood: the observation that previously described mutations affecting tetherin function do not alter structure suggests that sequence rather than structure is important. I suggest to rephrase these sentences to clarify this point.

(2) line 35: “developed numerous methods to evade viral Tetherin” should be rephrased as Tetherin is not viral.

(3) In Fig. 1, the C‐terminal anchor is modeled as an alpha-helix rather than a GPI anchor. The authors should clarify that there is evidence for both types of membrane anchor.

(4) Line 66: The authors state that “82 nucleotide polymorphisms” are listed in the Ensembl database. However, the total number of polymorphisms illustrated in Fig. 2A seems to be higher. What is the reason for this discrepancy?

(5) A few additional studies may be cited:
- lines 31/32: Discovery of the antiviral activity of tetherin: PMID 18200009, 18342597
- lines 36-38: Discovery of the signaling activity of tetherin: PMID 23159053
- lines 98/99: Lack of sequence conservation across species: PMID 26401043

(6) Lines 103/104: Why were the analyses limited to SNPs in the ectodomain?

(7) Lines 111-113: A study by Sakuma et al. suggests that dimerization is not essential for the antiviral activity of tetherin (PMID 19742323). This should be mentioned.

(8) Fig. 2: Do any of the missense mutations affect known glycosylation sites or disulfide bridges? An alignment illustrating SNPs and functional motifs may be helpful.
The color-coding may be misleading as blue and red illustrate synonymous/non-synonymous mutations in (A), but location on the outside/inside in (B).

(9) Fig. 3: The color-coding may again be misleading and suggest that the blue and red lines indicate the mean of mutations on the outside and inside of the dimer, respectively.

(10) Line 163: “over a region 18 amino acids in length”. The region seems to be even larger according to Fig. 5B.

(11) Lines 189-194: In their publication, Yang et al. suggest that “the clusters of charged residues, particularly those in the tetherin head domain, contribute to tetherin function, perhaps through opposing interactions with the negatively charged phospholipids of cellular and virion membranes or by maintaining specific structural properties of the tetherin head.” Even if the charged residues are not on a single face of the coiled-coil, these hypotheses may still be true. The authors should change their statement accordingly.

(12) Lines 234-236: The authors suggest that the proximity to glycosylation sites and a trafficking motif may explain the functional defects of the charge swap mutants. Are some of the SNPs also located in that region?

(13) Line 259: “most mutations had no effect on functions”. Which mutations had an effect on function? Were they also analyzed in the present study?

(14) Figure 4: Some of the data is covered by yellow boxes.

(15) Typos/minor corrections:
- line 39: “an N-terminal transmembrane helix”
- line 132: “(Figure 3)”
- line 139: “shown in Figure 4A”
- line 147: “(Figure 4B)”
- Fig. 4: “E for beta bridge” is not shown in the figure.
- Line 194: “Figure 7B-C”
- References/citations are not always in the correct format (e.g. line 236)

Reviewer 2 ·

Basic reporting

the manuscript is clear

Experimental design

experimental design is rational and relative complete

Validity of the findings

since most of the evidences are computational simulation, I suggest to add some experimental confirmation data.

Additional comments

Human Tetherin, also known as BST-2 plays important role in NF-kB signaling by binding enveloped viruses thus inhibiting viral release. The authors here aim to study the structure-function of Tetherin by screening several human genome mutations and estimate the effect on structure, they found that all the mutations did not affect the global structure of Tetherin but affected local structure changes. They also used the known functional defects to match their system, the conclusion is consistent. This work is interesting, but, for final publication, the manuscript needs some revisions.

Major:
1) Firstly, I suggest the author can pick up several mutants (Ala100Pro and Ile120Phe), which are not investigated before, and then do the functional assay to validate the conclusion. This will be more supportive for the paper publication.
2) The authors mentioned that the loss of the activity of the reversal of the charge in the N-terminal may not due to the proposed reason by Yang and coworkers. And the authors propose that this loss of activity may be because the mutation site is close to the trafficking region. Is there some data to support the hypothesis?

Minor:
1) in Figure 2A, residues 93-117 should be labeled, otherwise, it is hard to tell where this region locates;
2) the main text should be more carefully edited. For example, in line 231, malleability was mistakenly spelled as “malleablility” ; in line 256, the reference should be [Pace and Scholtz, 1998] instead of Pace and Scholtz [1998].

Annotated reviews are not available for download in order to protect the identity of reviewers who chose to remain anonymous.

---

## Round 0.2 · Minor Revisions

Please address remaining the minor critical points of reviewer #1 and amend your manuscript accordingly.

Reviewer 1 ·

Basic reporting

The manuscript is written in a clear and comprehensible manner. Professional English is used throughout (with still a few typos). The authors provide all background information that is required to understand the research question and the methods used. As suggested, additional literature has been cited in the revised version of the manuscript.

Experimental design

The research question is well defined. The methods are suitable to model the effects of SNPs on tetherin structure. One limitation of the present study is the lack of a cell biological assay analyzing the effects of the predicted structural changes on tetherin function. Although suggested by both reviewers, it still remains unclear whether mutations such as Ile120Phe affect the antiviral activity of tetherin by disrupting its (local) secondary structure.

Validity of the findings

The conclusions drawn by the authors are justified. The authors nicely discuss their findings in the context of previous publications that analyzed the structure-function relationship of tetherin.

Additional comments

In response to my previous comments, the authors have made several important changes that improved the quality of their manuscript. Among other things, they included two supplemental tables summarizing all mutations that were analyzed. Furthermore, they have cited additional studies and improved/corrected some of their figures.

Nevertheless, I encourage the authors to address a few remaining minor points:

1) Lines 213/214: The authors state that “the tetra-serine substitution showed large but localized changes in structure”. In Supplemental Table 2, however, local structural defects of this mutant are described with “NA”. Why?
2) Similarly, the authors highlight that “Arg64Pro and Ala100Pro induced alterations in secondary structure” (line 168). However, this defect is not indicated for the Arg64Pro mutant in Supplemental Table 1.
3) In line 167, the authors mention “five mutations” that disrupted helices. Which mutations are they referring to in this case?
4) Supplemental Table 1 lists 78 SNPs, while 77 are mentioned in the text (e.g. lines 69, 115, 119, …).
5) It still remains unclear whether the enrichment for missense mutations on the outside of tetherin is significant (see major point 2 of the initial review).
6) Lines 155/156: “Position 64 is the most frequently mutated position”. This is not entirely correct. Although five different missense mutations have been described at this position, they are very rare. Other positions are mutated more frequently.
7) Lines 163/165: Refer to Figure 4B.
8) Legend of Figure 7: “Overlap is shown in grey A yellow box indicates the area where these mutants are within the sequence.” The color of the box has been changed to blue.

Reviewer 2 ·

Basic reporting

the article is well organized

Experimental design

the experimental design is proper

Validity of the findings

because they basically focus on computational simulation, so they barely used the experimental data to support the conclusion

Additional comments

I still think it is important to pick up several mutants (Ala100Pro and Ile120Phe), which are not investigated before, and then do the functional assay to validate the conclusion.

---

## Round 0.3 · Minor Revisions

A) Figure 4B is not clear enough and some minor changes to its legend are needed. In fact, labeling the y-axis as "number of times observed as an outlier" is rather ambiguous: for example, the 4 amino acids surrounding 118 are coiled instead of helix in one of six simulations. Since one snapshot is taken every 25 ps and the simulations are 10 ns long, this implies that in 6 times 400 snapshots, about 400 snapshots correspond to outliers (i.e. coiled instead of the common alpha helix). Maybe the authors mean something else, such as "number of times in each simulation that the amino acids are outliers relative to the most frequent conformation IN THAT simulation". I think that this should be clarified.

B) The references to cowplot and ggridges should have working URL's

C) Authors seem to have used the full data of their simulations, instead of removing, as is usual practice, the initial portion as "equilibration run". Description of this procedure should be made explicit.

D) It would be great to have some more extensive analysis of why some mutations have longer-range effects. Since only one simulation of each of those mutants was performed, it is difficult to conclude that the observed changes are really due to the mutation rather than doe to some random simulation noise, unless a more detailed analysis of the non-replicated simulations shows that the changes do begin at the mutation site in the initial moments and then diffuse outwards from the mutation site. Ideally, additional runs should be performed to confirm that these results are real. Although, no additional simulation runs of those mutations are requested now, the authors should nonetheless discuss this (and/or eventually tone down some of their conclusions somehow).

---

## Round 0.4 · accepted · Accept

The authors should more carefully define outliers in fig 4b . It is currently stated : "Outliers were identified as sites that had distinct secondary structure more than two standard deviations above the average secondary structure value at that site across all simulations. A value of one means that the amino acid appeared as an outlier in 1 simulation." In the current plot, the y-axis runs from 0 to 84. However, the authors did not run such a large number of simulations. Therefore a value of 80 cannot mean that the amino acid appeared as an outlier in 80 simulations. Obviously, this should clarified. The corresponding clarification can be added during the proofing stage.